# Anthocyanins Protect Hepatocytes against CCl_4_-Induced Acute Liver Injury in Rats by Inhibiting Pro-inflammatory mediators, Polyamine Catabolism, Lipocalin-2, and Excessive Proliferation of Kupffer Cells

**DOI:** 10.3390/antiox8100451

**Published:** 2019-10-04

**Authors:** Dejan Popović, Gordana Kocić, Vuka Katić, Aleksandra Zarubica, Ljubinka Janković Veličković, Vanja P. Ničković, Andrija Jović, Andrej Veljković, Vladimir Petrović, Violeta Rakić, Zorica Jović, Nataša Poklar Ulrih, Danka Sokolović, Marko Stojanović, Marko Stanković, Goran Radenković, Gordana R. Nikolić, Аzra Lukač, Aleksandar Milosavljević, Dušan Sokolović

**Affiliations:** 1Department of Biochemistry, Faculty of Medicine, University of Niš, Bulevar dr Zorana Đinđića 81, 18000 Niš, Serbia; kocicrg@yahoo.co.uk (G.K.); veljkovicandrej@yahoo.com (A.V.); dusantsokolovic@gmail.com (D.S.); 2Department of Pathology, Faculty of Medicine, University of Niš, Bulevar dr Zorana Đinđića 81, 18000 Niš, Serbia; vuka.katic@gmail.com (V.K.); dravel@open.telekom.rs (L.J.V.); 3Department of Chemistry, Faculty of Science and Mathematics, University of Niš, Višegradska 33, 18000 Niš, Serbia; zarubica2000@yahoo.com; 4Clinical-Hospital Center Priština, 38205 Gračanica, Serbia; vanja.nickovic@gmail.com; 5Clinic of Skin and Venereal Diseases, Clinical Center of Niš, Bulevar dr Zorana Đinđića 48, 18000 Niš, Serbia; andrijajovic@rocketmail.com; 6Department of Histology and Embryology, Faculty of Medicine, University of Niš, Bulevar dr Zorana Đinđića 81, 18000 Niš, Serbia; vladimir.petrovic@medfak.ni.ac.rs (V.P.); radenkog@gmail.com (G.R.); 7College of Agriculture and Food Technology, Ćirila i Metodija 1, 18400 Prokuplje, Serbia; violetachem@gmail.com; 8Department of Pharmacology and Toxicology, Faculty of Medicine, University of Niš, Bulevar dr Zorana Đinđića 81, 18000 Niš, Serbia; profzoricajovic1953@gmail.com; 9Department of Food Science and Technology, Biotechnical Faculty, University of Ljubljana, Jamnikarjeva 101, 1000 Ljubljana, Slovenia; natasa.poklar@bf.uni-lj.si; 10Institute for Blood Transfusion in Nis, Bulevar dr Zorana Đinđića 48, 18000 Niš, Serbia; dankasokolovic@gmail.com; 11Faculty of Medicine, University of Niš, Bulevar dr Zorana Đinđića 81, 18000 Niš, Serbia; mimtroska@gmail.com (M.S.); marko.stankovic018@gmail.com (M.S.); a.milosavljevic993@gmail.com (A.M.); 12Medical Faculty, University of Priština, 38220 Kosovska Mitrovica, Serbia; gordananikolic65@yahoo.com; 13Health Center Rožaje, 84310 Rožaje, Montenegro; azra430@hotmail.com

**Keywords:** carbon tetrachloride, anthocyanins, anti-inflammatory effects, kupffer cells, lipocalin-2, polyamine catabolism

## Abstract

This study examined the hepatoprotective and anti-inflammatory effects of anthocyanins from *Vaccinim myrtillus* (bilberry) fruit extract on the acute liver failure caused by carbon tetrachloride-CCl_4_ (3 mL/kg, i.p.). The preventive treatment of the bilberry extract (200 mg anthocyanins/kg, orally, 7 days) prior to the exposure to the CCl_4_ resulted in an evident decrease in markers of liver damage (glutamate dehydrogenase, sorbitol dehydrogenase, malate dehydrogenase), and reduced pro-oxidative (conjugated dienes, lipid hydroperoxide, thiobarbituric acid reactive substances, advanced oxidation protein products, NADPH oxidase, hydrogen peroxide, oxidized glutathione), and pro-inflammatory markers (tumor necrosis factor-alpha, interleukin-6, nitrite, myeloperoxidase, inducible nitric oxide synthase, cyclooxygenase-2, CD68, lipocalin-2), and also caused a significant decrease in the dissipation of the liver antioxidative defence capacities (reduced glutathione, glutathione S-transferase, and quinone reductase) in comparison to the results detected in the animals treated with CCl_4_ exclusively. The administration of the anthocyanins prevented the arginine metabolism’s diversion towards the citrulline, decreased the catabolism of polyamines (the activity of putrescine oxidase and spermine oxidase), and significantly reduced the excessive activation and hyperplasia of the Kupffer cells. There was also an absence of necrosis, in regard to the toxic effect of CCl_4_ alone. The hepatoprotective mechanisms of bilberry extract are based on the inhibition of pro-oxidative mediators, strong anti-inflammatory properties, inducing of hepatic phase II antioxidant enzymes (glutathione S-transferase, quinone reductase) and reduced glutathione, hypoplasia of Kupffer cells, and a decrease in the catabolism of polyamines.

## 1. Introduction

Carbon tetrachloride (CCl_4_), allyl alcohol, 1-naphthyl isocyanate, and thioacetamide are hepatotoxic substances which are used in experimental medicine as proven models of acute liver damage with necrotic changes in various zones of the liver lobules [1,2]. One of the most well-known and well-used experimental models of acute liver injury is CCl_4_ [3]. This model of chemical liver damage is used for the examination of the liver damage mechanisms and of the possible anti-hepatotoxic (hepatoprotective) activities of various synthetically generated substances or natural products [4,5]. The liver damage caused by CCl_4_ is similar to human liver disorders, judging by its morphological and biochemical characteristics [5]. Molecular CCl_4_ is not toxic, but its hepatotoxicity develops after metabolic activation in the liver and the formation of the highly reactive trichloromethyl (^●^CCl_3_) and trichloromethyl-peroxide of the radicals (CCl_3_O_2_^●^) [4,6]. Toxic metabolites, CCl_4_ are tied to the lipids, and they remove the hydrogen atom from the unsaturated fat acids of the membrane, which induces the process of chain lipid peroxidation that damages the liver cells [3].

There exist over 600 commercial hepatoprotective diet supplements worldwide, which contain approximately 100 various medical plants [7]. Bilberry (Vaccinium myrtillus) is a shrub-like perennial plant growing in the mountains of Europe and North America. Its berries are purple or dark-blue in colour, and represent one of the richest sources of phenolic components. The blue colour of the berries originates from the anthocyanins, the predominant component of the bilberry fruit [8,9]. The bilberry contains various phenolic components (anthocyanins, flavanols, flavonols, phenol acids), sugars, pectines, and vitamins. Anthocyanins are composed of anthocyanidins and sugar components [10]. Numerous studies have proven the anti-oxidative, proapoptotic, antimicrobial, and anti-cancer effects of anthocyanins and other active bilberry components [9,10]. Anthocyanins have the ability to neutralize free radicals, prevent the process of lipid peroxidation, and regulate the release of pro-inflammatory mediators [11].

Acute liver damage is the basis of the origin and development of numerous liver disorders that can lead to its terminal failure. Frequent liver damage provokes fibrosis, cirrhosis, and hepatocellular carcinoma. In recent years, the acute liver damage caused by the pharmacotherapeutic protocols, drug overdose, self-medication, alcohol abuse, and viral hepatitis infection has been increasingly present [12,13]. The initial response to acute damage is an increase in pro-inflammatory mediators that promote hepatoprotection and liver regeneration [13]. However, inappropriately controlled increase in pro-inflammatory mediators (TNF-α, IL-6, Lipocalin-2) and activation of Kupffer cells, as well as increased catabolism of the polyamine with the release of the cytotoxic H_2_O_2_, can significantly aggravate the liver damage present [13,14,15].

The purpose of this study was to investigate the anti-inflammatory and hepatoprotective properties of the bilberry fruits’anthocyanins by assessing the biochemical parameters of liver damage, various pro-oxidative, pro-inflammatory, and antioxidative markers, as well as histopathological, morphometric, and immunohistochemical analyses associated with inflammation and liver cell damage.

## 2. Materials and Methods

### 2.1. Chemicals

Ammonium chloride (NH_4_Cl) > 99% (Centrohem, Stara Pazova, Serbia); Ammonium iron (II) sulfate hexahydrate ((NH_4_)_2_Fe(SO_4_)_2_ x 6H_2_O) ≥ 98% (Sigma-Aldrich, St. Louis, MO, USA); L-Arginine ≥ 98% (Sigma-Aldrich, St. Louis, MO, USA); 2,2′-Azino-bis(3-ethylbenzothiazoline-6-sulfonic acid) diammonium salt (ABTS) ≥ 98% (Sigma-Aldrich, St. Louis, MO, USA); Ascorbic acid 99% (Acros Organic, Geel, Belgium); Bovine serum albumin (BSA); lyophilized powder ≥ 96% (Sigma-Aldrich, St. Louis, MO, USA); 2,3-Butanedionemonoxime (BDM) ≥ 98% (Sigma-Aldrich, St. Louis, MO, USA); 1-Chloro-2,4dinitrobenzene (CDNB) ≥ 99% (Sigma-Aldrich, St. Louis, MO, USA); Chloramine T trihydrate 98% (Sigma-Aldrich, St. Louis, MO, USA); Chloroform (CHCl_3_) > 99% (Centrohem, Stara Pazova, Serbia); Carbon tetrachloride (CCl_4_) > 99% (Riedel-de Haën, Seelze, Germany); Ciocalteus phenol reagent (Sigma-Aldrich, St. Louis, MO, USA); Copper (II) sulfate anhydrous [CuSO_4_] > 99% (Sigma-Aldrich, St. Louis, MO, USA); Cyclohexane > 99.9% (Centrohem, Stara Pazova, Serbia); Dextrose ≥ 99.5% (Merck, Kenilworth, NJ, USA); 2,6-Dichlorophenolindophenol sodium salt hydrate (DCPIP) ≥ 98% (CARL ROTH, Karlsruhe, Germany); 3,3′-Dimethoxybenzidine (o-Dianisidine) 97% (Acros Organic, Geel, Belgium); 5,5′-dithiobis-(2-nitrobenzoic acid) [DTNB] 99% (Acros Organics, Geel, Belgium); Flavine-adenine-dinucleotide disodium salt (FAD) ≥ 95%, for biochemistry, (CARL ROTH, Karlsruhe, Germany); Fructose ≥ 99% (Sigma-Aldrich, St. Louis, MO, USA); Glacial acetic acid (CH_3_COOH) ≥ 99% (Centrohem, Stara Pazova, Serbia); L-Glutathione oxidized (GSSG) 98% (Acros Organic, Geel, Belgium); L-Glutathione reduced (GSH) ≥ 98.0% (Sigma-Aldrich, St. Louis, MO, USA); Glutathione reductase (GR) from baker’s yeast (Sigma-Aldrich, St. Louis, MO, USA); Hydrochloric acid (HCl) ≥ 37% (Centrohem, Stara Pazova, Serbia); Horseradish peroxidase 85 U/mg dry weight (Alfa Aeser, Haverhill, Massachusetts, USA); H_2_O_2_ 30% (Carlo erba, Val-de-Reuil, France); Iron(II) sulfate heptahydrate (FeSO_4_ x 7H_2_O) ≥ 99% (Sigma-Aldrich, St. Louis, MO, USA); Ferric chloride-Iron(III) chloride hexahydrate (FeCl_3_ x 6H_2_O) ≥ 99% (Sigma-Aldrich, St. Louis, MO, USA); Ketamidor (Richter Pharma AG, Wels, Austria); α-Ketoglutaric acid > 99% (Sigma-Aldrich, St. Louis, MO, USA); Magnesium chloride hexahydrate (MgCl_2_ x 6H_2_O) > 98% (Centrohem, Stara Pazova, Serbia); L-Malic acid 97% (Sigma-Aldrich, St. Louis, MO, USA); Manganese (II) chloride [MnCl_2_] ≥ 99% (Sigma-Aldrich, St. Louis, MO, USA); Methanol (CH_3_OH) for HPLC ≥ 99.9% (Sigma-Aldrich, St. Louis, MO, USA); 2-Methoxyphenol (Guaiacol) ≥ 98% (Alfa Aesar, Haverhill, Massachusetts, USA); 3-methyl-2-benzothiazolinone hydrazone hydrochloride hydrate 97% (Sigma-Aldrich, St. Louis, MO, USA); Na_2_EDTA dihydrate (Calbiochem, San Diego, California, USA); β-Nicotinamide adenine dinucleotide phosphate tetrasodium salt (NADPH-Na_4_) ≥ 95%, for biochemistry, (CARL ROTH, Karlsruhe, Germany); β-Nicotinamide adenine dinucleotide, disodium salt, reduced, hydrate (NADH) ≥ 95% (Acros Organic, Geel, Belgium); β-Nicotinamide adenine dinucleotide phosphate disodium salt (NADP-Na_2_) ≥ 97% (Sigma-Aldrich, St. Louis, MO, USA); Ninhydrin, ACS reagent (Sigma-Aldrich, St. Louis, MO, USA); N-(1-naphthyl) ethylenediamine dihydrochloride (NEDA) > 98% (Sigma-Aldrich, St. Louis, MO, USA); PBS-Phosphate buffered saline, tablet, (Sigma-Aldrich, St. Louis, MO, USA); Phenol red ≥ 98% (Acros Organic, Geel, Belgium); Dibasic potassium phosphate (K_2_HPO_4_) > 98% (Centrohem, Stara Pazova, Serbia); Potassium dihydrogen phosphate (KH_2_PO_4_) > 98% (Centrohem, Stara Pazova, Serbia); Potassium sodium tartrate tetrahydrate > 99% (Centrohem, Stara Pazova, Serbia); Putrescine dihydrochloride ≥ 98% (Sigma-Aldrich, St. Louis, MO, USA); Sodium carbonate anhydrous (Na_2_CO_3_) > 99% (Centrohem, Stara Pazova, Serbia); Sodium chloride (NaCl) > 99% (Merck, Kenilworth, NJ, USA); Sodium hydroxide (NaOH) > 98% (Centrohem, Stara Pazova, Serbia); Sulfuric acid (H_2_SO_4_) 95–97% (Centrohem, Stara Pazova, Serbia); Spermine ≥ 97% (Sigma-Aldrich, St. Louis, MO, USA); Sodium 5,5-diethylbarbiturate-Barbitone (Sigma-Aldrich, St. Louis, MO, USA); Sulfanilamide ≥ 99% (Sigma-Aldrich, St. Louis, MO, USA); 2-Thiobarbituric acid (TBA) ≥ 98% (Sigma-Aldrich, St. Louis, MO, USA); Thiosemicarbazide ≥ 99% (Sigma-Aldrich, St. Louis, MO, USA); Trichloroacetic acid (CCl_3_COOH) ≥ 99% (Sigma-Aldrich, St. Louis, MO, USA); Trizma^®^ base ≥ 99.9% (Sigma-Aldrich, St. Louis, MO, USA); Triethanolamine (TEA) 99% (Centrohem, Stara Pazova, Serbia); TWEEN^®^ 20 (Sigma-Aldrich, St. Louis, MO, USA); Urease from Canavalia ensiformis (Jack bean), Type III, powder, 15,000-50,000 units/g solid, (Sigma-Aldrich, St. Louis, MO, USA); 2-Vinyl pyridine 97% (Sigma-Aldrich, St. Louis, MO, USA); Xylenol orange (Centrohem, Stara Pazova, Serbia); 2,3-Bis(2-methoxy-4-nitro-5-sulfophenyl)-2*H*-tetrazolium-5-carboxanilide inner salt (XTT) ≥ 90% (Sigma-Aldrich, St. Louis, MO, USA); Zinc sulfate heptahydrate (ZnSO_4_ x 7H_2_O) > 99% (Centrohem, Stara Pazova, Serbia).

### 2.2. Bilberry Fruit Extract Preparation and HPLC Analysis

In this study, bilberry (*Vaccinium myrtillus*) extract was obtained by the extraction from fully ripe bilberry fruits harvested in the forests of Slovenia, according to the protocol and procedure previously described by Može et al., (2011) [9]. The extraction process was performed using the ice-cold deoxygenated methanol (CH_3_OH) in a ratio of 3:1 to bilberry fruits. The process of identification of the bilberry plant species and fruit, as well as the extraction procedure were performed by Prof. Ulrich at the Department of Food Science and Technology, Biotechnical Faculty, Slovenia. Toxic methanol from the bilberry fruit extract (2000 mL) was eliminated by evaporation on a rotary evaporator (RV 8V, IKA, Staufen, Germany) under reduced pressure at 40°C to obtain a semi-liquid concentrated extract (120 mL). The semi-liquid concentrated extract was used in an in vivo part of the experiment of this study. Anthocyanins from the bilberry fruit extract were determined by the liquid chromatography–mass spectrometry/mass spectrometry (LC-MS/MS) according to a procedure previously described by Može et al., (2011) [9].

### 2.3. Animals and Protocol

The experiment was conducted on adult male Wistar rats, weighing 310 ± 20 g and about 180 days old (6 months), obtained from Vivarium at the Faculty of Medicine in Niš, Serbia. The animals were kept in strictly controlled, standard, and sterile conditions (temperature 20 ± 2 °C, relative humidity 50 ± 10%, and light/dark cycle at 12/12h), with free access to tap water and standard laboratory rat pellets. The study involved 32 rats, divided into four equal groups, previously adapted to the experimental conditions. Each group contained eight laboratory rats housed in polyacrylic cages. All experimental procedures with the laboratory animals were performed according to the National Institutes of Health guide (NIH) and permission was obtained from the Ethics Committee of the Faculty of Medicine in Niš (01-9337-16). The experimental procedures were performed at the Vivarium of the Biomedical Center and the laboratories of the biochemical department at the Faculty of Medicine in Niš.

### 2.4. Experimental Groups 

**Group I** (**untreated**
**group**) was administered the physiological solution (0.9% NaCl) in single doses of 0.83 ± 0.05 mL orally once a day, for 7 days. On the seventh day, 3 h after the physiological solution had been administered, the animals received corn oil in a dose of 6 mL/kg b.w., intraperitoneally (i.p.).

**Group II** (**treated group**) was administered bilberrry anthocyanins (0.83 ± 0.05 mL), in single doses of 200 mg of the anthocyanins/kg b.w. orally once a day, for 7 days. On the seventh day, 3 h after the last dose of the bilberry fruit extract, the animals received corn oil in a dose of 6 mL/kg b.w., i.p. The rats received bilberry extract whose dose was calculated on the basis of the presence of anthocyanins in the semi-liquid concentrated extract [16].

**Group III** (**CCl_4_ + vehicle treated group**) was administered the physiological solution in single doses of 0.83 ± 0.05 mL orally once a day, for 7 days. On the seventh day, 3 h after the physiological solution had been administered, the animals received CCl_4_ solution in a dose of 6 mL/kg b.w., i.p. CCl_4_ and vehicle corn oil were mixed in a 1:1 (v:v) ratio [3 mL/kg b.w. CCl_4_ + 3 mL/kg b.w. corn oil] [16]. The occurrence of hemorrhagic necrotic fields (anuclear eosinophilic mass in the liver lobes) and severe degenerative changes (macrovesicular hepatocytes), followed by an increase of biochemical markers of liver damage (GLDH and SDH) were the main criteria for CCl_4_ dose selection.

**Group IV (treated group + CCl_4_)** was administered anthocyanins from the *Vaccinium myrtillus* (bilberry) extract in single doses of 200 mg of the anthocyanins/b.w. (0.83 ± 0.05 mL) orally once a day, for 7 days, and on the seventh day, 3 h after the last dose of the anthocyanins, a single dose of CCl_4_ solution (6 mL/kg b.w., i.p.) was administered.

### 2.5. Sample Preparation for Biochemical Analyses

All animals were sacrificed 24 h after the CCl_4_ administration. The rats were anesthetized using Ketamidor (2 mL/kg b.w., i.p.). The blood samples obtained by cardiac puncture were centrifuged at 3000 rpm for 10 min at 4 °C, and the serum obtained was stored at −20 °C until the biochemical analysis was performed. The liver homogenate was obtained by grinding the liver tissue (homogenization) in PBS buffer (pH = 7.4, in cold water at 4 °C) using a homogenizer (T-18 Ultra Turrex, IKA, Staufen, Germany). The liver homogenate (10% *w*/*v*) was centrifuged at 3000 rpm for 15 min at 4 °C in order to eliminate non-homogenized and dead cells, after which a post-mitochondrial liver supernatant was further centrifuged at 10,000 rpm for 30 min at 4 °C. The post-mitochondrial liver supernatant was used to determine the total oxidative status, total antioxidant capacity, advanced oxidation protein products, thiobarbituric acid reactive substances, hydrogen peroxide, NADPH oxidase, nitrite, arginase, citrulline, putrescine oxidase, and spermine oxidase.

### 2.6. Determination of Glutamate Dehydrogenase (GLDH) Activity 

The activity of GLDH was measured from the reaction mixture contained: 2.5 mL of the Tris buffer (100 mM, pH = 8.3), 0.2 mL of the NH_4_Cl (3.3 M), 0.1 mL of the α-ketoglutaric acid (225 mM, pH = 8.3), 0.1 mL of the NADPH (7.5 mM), and 0.1 mL of the serum [17]. The decrease in absorbance was recorded spectrophotometrically at 340 nm. The activity of GLDH has been expressed in U/L.

### 2.7. Determination of Sorbitol Dehydrogenase (SDH) Activity

The activity of SDH was measured from the reaction mixture contained: 0.7 mL of the NADH (355 μM, dissolved in 100 mM of the Tris buffer, pH = 6.6), 0.1 mL of the serum, and 0.2 mL of the fructose (2.5 M, dissolved in the Tris buffer, pH = 6.6) [18]. The changes in absorbance were recorded spectrophotometrically at 340 nm. The activity of SDH has been expressed in U/L.

### 2.8. Determination of Malate Dehydrogenase (MDH) Activity

The activity of MDH was determined from the experimental mixtures contained: 0.895 mL of the Tris buffer (50 mM, pH = 7.8), 0.05 mL of the L-Malic acid (0.2 M), 0.025 mL of the NADP (0.01 M), 0.01 mL of the MnCl_2_ (0.1 M), and 0.02 mL of the serum [19]. The increase in the absorbance was read at 340 nm. The activity of MDH has been expressed in U/L.

### 2.9. Determination of (TOS) Total Oxidative Status

The concentration of the TOS was determined according to the Erel, (2005) method [20]. The reaction mixture contained: 225 μL of the Reagent I (140 mM NaCl, 150 μM xylenol orange, and 1.35 M glycerol dissolved in 25 mM H_2_SO_4_, pH = 1.75), 35 μL of the liver homogenate, and 11 μL of the reagent II (10 mM o-dianisidine and 5 mM (NH_4_)_2_Fe(SO_4_)_2_ dissolved in 25 mM H_2_SO_4_). After 3 min the absorbance was recorded spectrophotometrically at 560 nm, and the results are expressed as nM/mg protein.

### 2.10. Determination of (TAC) Total Antioxidant Capacity

The concentration of the TAC was determined according to the Erel (2004) method [21]. The reaction mixture contained: 200 μL of the reagent I (acetate buffer-CH_3_COONa/CH_3_COOH, 0.4 M, pH = 5.8), 20 μL of the liver homogenate, and 20 μL of the reagent II (ABTS 10 mM, 2 mM H_2_O_2_ in acetate buffer (30 mM), pH = 3.6). After 5 min the absorbance was recorded spectrophotometrically at 660 nm, and the results are expressed as nM/mg protein.

### 2.11. Determination of Conjugated Dienes (CD) Concentration

The conjugated dienes (CD) were determined according to the method by Devasagayam et al., (2003) [22]. The non-centrifuged liver homogenate (0.5 mL) was vigorously treated with 5 mL of a solvent mixture-CHCl_3_:CH_3_OH = 2:1. After the centrifugation at 3000 rpm for 5 min, 2 mL of the lower layer was taken and evaporated to dryness in a nitrogen stream at 45 °C. The evaporated content was dissolved in 1 mL of the cyclohexane. The absorbance was read at 233 nm against the cyclohexane. The concentration of conjugated dienes were expressed as nM/mg liver proteins.

### 2.12. Determination of Lipid Hydroperoxide (LOOH) Concentration

The concentration of the lipid hydroperoxide was measured by the spectrophotometric method of Lima et al., (1995) [23]. The reaction assay contained: 0.25 mL of the FeSO_4_ (1 mM), 0.1 mL of the H_2_SO_4_ (0.2 M), 0.1 mL of the xylenol orange (1 mM), 0.45 mL of the distilled water, and 0.1 mL of the diluted methanol extract of the liver homogenate (1:5). After the incubation of 120 min at room temperature, the first absorbance was read at 580 nm, and the second after the addition of 0.005 mL of the cumene hydroperoxide (5 mM) and the 30 min incubation. The concentration of the LOOH is expressed as nM/mg protein of the liver tissue.

### 2.13. Determination of Thiobarbituric Acid Reactive Substances (TBARS) Concentration

The TBARS concentration was measured according to a slightly modified method previously described by Sahreen et al., (2011) [24]. The TBARS method was used to determine a low molecular weight of the aldehyde-malondialdehyde (MDA) that reacts with the thiobarbituric acid, forming a pink complex. The reaction mixture contained: 0.2 mL of the liver homogenate, 0.2 mL of the ascorbic acid (100 mM), 0.58 mL of the potassium phosphate buffer (0.1 M; pH = 7.4), and 0.02 mL of the Ferric chloride-FeCl_3_ (100 mM). After being incubated for 60 min in a water bath at 37 °C, 1 mL of the CCl_3_COOH (10%) was added to the reaction assay. Then, 1 mL of the TBA (0.67% dissolved in 0.1 M NaOH) was added to the reaction assay and heated for 30 min in a boiling water bath (100 °C), and after cooling, 5 mL of the butanol-pyridine mixture (15:1) was added to the reaction mixture. After the centrifugation at 4000 rpm for 10 min, the absorbance of a clear supernatant layer was read at 535 nm against a butanol-pyridine mixture. The concentration of the TBARS has been expressed in nM/mg liver proteins.

### 2.14. Determination of Advanced Oxidation Protein Products (AOPP) Concentration

The concentration of the AOPP measured from the reaction mixture contained: 10 μL of the potassium iodide-KI (1.16 M), 200 μL of the diluted liver homogenate, and 20 μL of the glacial CH_3_COOH [25]. Immediately after the glacial acetic acid was added, the absorbance was read at 340 nm against blank. The concentration of the AOPP was calculated based on the standard curve of chloramine T (0–100 μmol/L) and expressed in nM/mg protein.

### 2.15. Determination of Hydrogen Peroxide (H_2_O_2_) Concentration

The concentration of H_2_O_2_ determined from the reaction mixture contained: 2 mL of the liver homogenate and 1 mL of the phenol red solution (5.5 mM glucose, 8.5 U/mL horseradish peroxidase, 0.28 mM phenol red, and 140 mM NaCl solved in 10 mM Potassium Phosphate Buffer, pH = 7.0) [26]. The reaction assay was incubated for 60 min in a water bath at 37 °C. The reaction was stopped by the addition of 10 μL of the NaOH (10 M), and then centrifuged at 4000 rpm for 10 min. The absorbance of the supernatant was read at 610 nm against blank. The H_2_O_2_ concentration was expressed as nM H_2_O_2_/mg of liver proteins based on a standard curve of H_2_O_2_ which oxidized phenol red.

### 2.16. Determination of NADPH Oxidase Activity

The activity of NADPH oxidase was determined according to the method by Jiang and Zhang, (2002) [27]. The reaction mixture contained 12.5 μL of the Tris buffer (1 M, pH = 7.5), 125 μL of the XTT (1 mM), 25 μL of the NADPH (1 mM), 77.5 μL of the distilled water, and 10μL of the liver homogenate. The changes in the absorbance were recorded at intervals of 20 min and read at 340 nm against the control. The activity of NADPH oxidase has been expressed in nM/mg protein.

### 2.17. Determination of Oxidized Glutathione (GSSG) Concentration

The concentration of the oxidized glutathione-GSSG was measured according to a method previously described by Rahman et al., (2006) [28]. The reaction assay contained: 100 μL of liver homogenate (diluted) mixed with 2 μL of 2-vinylpyridine. The assay was incubated with stirring for 60 min, and then 6 μL of triethanolamine, 60 μL DTNB (2 mg/3 mL), 60 μL GR (10 U/3 mL), and 60 μL NADPH (2 mg/3 mL) were added to the reaction assay. The changes in absorbance were recorded at 412 nm. The concentration of the oxidized glutathione has been expressed in nM/mg protein and obtained from the standard curve of GSSG.

### 2.18. Determination of Nitrite (NO_2_^−^) Determination

The concentration of NO_2_^-^ was determined according to the method by Sajid et al., (2016) [29]. The liver homogenate (0.1 mL) was treated with 0.1 mL of the ZnSO_4_ (5%) and 0.1 mL of the NaOH (0.3 M). After the centrifugation at 6400× *g* for 15 min at 4 °C, 20 μL of the supernatant was added in 1 mL of the Greiss reagent. The absorbance was recorded at 540 nm against blank. The concentration of NO_2_^−^ was expressed as μM/mg protein based on a standard curve of Sodium nitrite-NaNO_2_.

### 2.19. Determination of Myeloperoxidase (MPO) Activity

The activity of myeloperoxidase measured from the reaction assay contained: 3 mL of 0.013 M 2-methoxyphenol and 0.00033 M H_2_O_2_ (dissolved in the potassium phosphate buffer 0.01 M, pH = 7.0) [30]. After adding 35 μL of the liver homogenate, changes in the absorbance were read at 470 nm against blank. The activity of myeloperoxidase (MPO) has been expressed in U/mg proteins.

### 2.20. Determination of Arginase Activity

The activity of arginase measured from the reaction assay contained: 0.1 mL of the liver homogenate, 0.4 mL of the distilled water, 0.1 mL of the arginine (200 mM), 0.1 mL of the MnCl_2_ (50 mM), and 0.5 mL of the sodium barbitone buffers (0.1 M, pH = 9.5) [31]. After a 30 min incubation at 37 °C, the reaction was stopped by the addition of 1.3 mL of CCl_3_COOH (20%). The reaction mixture was centrifuged for 10 min at 3000 rpm, and then 0.1 mL of the supernatant was mixed with 0.9 mL of CCl_3_COOH (10%), 1 mL of the glacial CH_3_COOH and 0.5 mL of the ninhydrin reagent (250 mg of ninhydrin was dissolved in a mixture of 4 mL of 6 M H_3_PO_4_, and 6 mL glacial CH_3_COOH). The reaction assay was heated for 30 min at 95 °C and, after cooling, the absorbance was read at 515 nm against the control. The enzyme activity has been expressed in μmol/mg of protein.

### 2.21. Determination of Citrulline Concentration

The concentration of citrulline was measured according to the method previously described by Knipp and Vašák, (2000) [32]. The reaction mixture consisted of 200 μL of the reagent I (reagents A and B mixed in a ratio of 1:3), 20 μL of the arginine (10 mM), and 40 μL of the liver homogenate. Reagent A contained 80 mM of the diacetylmonoxime and 2 mM of the thiosemicarbazide dissolved in distilled water, and Reagent B contained 3 M H_2_SO_4_, 6 M H_3_PO_4_, and 2 mM NH_4_Fe(SO_4_)_2_ dissolved in distilled water. The experimental mixture was heated at 95 °C for 15 min. After cooling, the absorbance was read at 530 nm against the control. The citrulline concentration has been expressed in μmol/mg of protein.

### 2.22. Determination of Putrescine Oxidase (PutOX) Activity

The activity of the PutOX was determined according to a method described by Quash et al., (1972) [33]. The reaction assay contained 0.02 mL of the liver homogenate, 0.1 mL of the NaCl-Tris HCl buffer (4 mM Tris, 0.14 M NaCl, pH = 7.7), and 0.1 mL of the putrescine (18.3 mM dissolved in NaCl-Tris buffer). The reaction assay was incubated for 60 min at 37 °C in a water bath. After the addition of 0.1 mL of 0.4% 3-methyl-2-benzothiazolinone hydrazone hydrochloride hydrate, the mixture was incubated for 30 min at 25 °C, and thereafter, 0.5 mL of FeCl_3_ was added to the mixture. After 15 min, the absorbance was read at 660 nm against the control. The activity of PutOX has been expressed in nM/mg of liver protein.

### 2.23. Determination of Spermine Oxidase (SpmOX) Activity

The activity of the SpmOX was determined according to a method described by Quash et al., (1972) [33]. The reaction mixture contained 0.02 mL of the liver homogenate, 0.1 mL of the NaCl-Tris HCl buffer (10 mM Tris, 0.14 M NaCl, pH = 7.2), and 0.1 mL of the spermine (17.2 mM dissolved in NaCl-Tris buffer). The reaction mixture was incubated for 4 h at 37 °C in a water bath. After the addition of 0.1 mL of 0.4% 3-methyl-2-benzothiazolinone hydrazone hydrochloride hydrate, the mixture was incubated for 30 min at 25 °C, and thereafter, 0.5 mL of FeCl_3_ was added to the mixture. After 15 min, the absorbance was read at 660 nm against the control. The activity of SpmOX has been expressed in nM/mg of liver protein.

### 2.24. Determination of Reduced Glutathione (GSH) Concentration

The concentration of the total glutathione and reduced glutathione (GSH) were measured according to a method previously described by Rahman et al., (2006) [28]. The experimental mixture for determining the total glutathione contained: 20 μL of the liver homogenate (diluted), 60 μL of the DTNB (2 mg/ 3 mL), 60 μL of the GR (10 U/ 3mL), and 60 μL of the NADPH (2 mg/3 mL). The changes in the absorbance were read spectrophotometrically at 412 nm. The concentration of the reduced glutathione (GSH) was calculated by subtracting the values from the total glutathione and oxidized glutathione (GSSG), and expressed in nM/mg proteins.

### 2.25. Determination of Glutathione S-transferase (GST) Activity

The activity of GST measured from the reaction mixture contained: 2.7 mL of the potassium phosphate buffer (0.1 M, pH = 6.5; 1 mM EDTA), 0.1 mL GSH (75 mM), 0.1 mL of the CDNB (30 mM), and 0.1 mL of liver homogenate (diluted) [34]. The changes in the absorbance were determined spectrophotometrically at 340 nm. The activity of GST has been expressed in nM/mg of liver protein.

### 2.26. Determination of Quinone Reductase (QR) Activity

The activity of QR was determined according to a method described by Shah et al., (2015) [35]. The reaction mixture contained of 2.06 mL of the Tris buffer (0.025 M, pH 7.4), 0.02 mL of the Tween-20 (1% *w/v*), 0.1 mL of the FAD (150 μM), 0.02 mL of the NADPH (0.1 mM), 0.7 mL of the BSA (1 mg/mL), 0.05 mL of liver homogenate (10%), and 0.05 mL of the 2,6-dichlorophenolindophenol (2.4 mM). The changes in the absorbance were recorded at 600 nm. The activity of QR has been expressed in nM/mg of liver protein.

### 2.27. Determination of Lipocalin-2 (NGAL, LCN2), CD68, TNF-α, IL-4, IL-6, and IL-13 Concentration

The liver homogenate concentration level of Lipocalin-2 (Neutrophil gelatinase-associated lipocalin), CD68 (Cluster of Differentiation 68), TNF-α (tumor necrosis factor-alpha), IL-4 (interleukin-4), IL-6 (interleukin-6), and IL-13 (interleukin-13) were determined using a commercial ELISA kit (Rat NGAL ELISA kit, ab207925, Abcam, USA; Rat CD68 ELISA kit, CSB-E13297r, Cusabio, USA; Rat TNF-α ELISA kit, Antibodies-online, Germany; Rat IL-4 ELISA kit, Antibodies-online, Germany; Rat IL-6 ELISA kit, R&D Systems, USA; Rat IL-13 ELISA kit, Antibodies-online, Germany) according to the manufacturer’s instructions. Lipocalin-2, CD68, TNF-α, IL-4, IL-6, and IL-13 were determined from a standard curve. The concentrations of NGAL, TNF-α, IL-6, and IL-13 have been expressed in pg/mg liver protein, while the concentration of CD68 and IL-4 have been expressed ng/mg liver proteins.

### 2.28. Determination of Protein Concentration

The protein concentration in the liver homogenate was determined according to the method previously described by Popović et al., (2019) [16]. The reaction mixture contained 10 μL of the diluted liver homogenate and 150 μL of the reagent C (1 mL of 1% CuSO_4_, 1 mL of 2% potassium sodium tartrate, and 98 mL Na_2_CO_3_ dissolved in 0.1 M NaOH). After 30 min of incubation, 30 μL of the Folin reagent was added to the mixture (Folin–Ciocalteu reagent and water mixed in a ratio of 1:2). After 20 min, the absorbance was read at 550 nm, and the proteins concentration expressed as mg protein/L.

### 2.29. Histopathological Examinations and Morphometric Analysis

A slice of the liver tissue from the left lobule was fixed in 10% neutral phosphate-buffered formalin for at least 24 h and then embedded in paraffin after the tissue-processing process had been done according to the standard protocol provided for laboratory animals [16]. Cross-sections of the liver tissue were made on a Leica microtome (Leica Microsystems, Germany) with a thickness of 5 μm, and then deparaffinized, hydrated, and stained with hematoxylin and eosin (H&E) protocol. Pathologists have analyzed in great detail the changes (the presence of necrosis, hyperplasia, and hypertrophy of Kupffer cells, vacuolar, hydropic, and fatty hepatocytes, as well as inflammatory infiltrate) in all three zones of the liver lobule for each animal individually in each group. The morphometric analysis was used to determine the extent of the necrotic area on a selected H&E field. For each animal in each group, 10 microscopic fields at 200× magnification were examined by morphometric analysis. An Olympus BX-50 microscope (Shinjuku, Tokyo, Japan) with a Leica digital camera (DFC295, Reuil-Malmaison, France) was used to obtain photomicrographs.

### 2.30. Immunohistochemical Analysis

The liver tissue sections were deparaffinized in xylene, hydrated with a series of decreasing concentrations of ethanol, treated with 0.3% H_2_O_2_-CH_3_OH solution for 10 min, heated in 10 mM citrate buffer (pH = 6.0) at 121°C for 30 min, and subsequently blocked with 5% BSA in 0.01 M phosphate buffer. The sections were incubated overnight at 4 °C together with the primary antibody. After incubation with the secondary antibody and visualization with 3,3′-diaminobenzidine (DAB), the liver tissue sections were stained with hematoxylin and ready for the immunohistochemical analysis. The following primary antibodies were purchased from Santa Cruz Biotechnology (Santa Cruz, California, USA) and used in this study: COX-2 (cyclooxygenase-2) antibody (sc-19999), iNOS (inducible nitric oxide synthase) antibody (sc-7271), TNF-α antibody (sc-52746), NGAL antibody (sc-515876), and the CD68 antibody (sc-20060), applied in dilutions of 1:200, 1:100, 1:100, 1:200, and 1:100, respectively.

### 2.31. Statistical Analysis

All statistical analyses were examined using a one-way analysis of variance (ANOVA) test followed by post hoc Tukey’s multiple comparison tests. The results have been expressed as the mean ± standard error (SD). Statistical significance was based on *p* ≤ 0.05.

## 3. Results

### 3.1. Bilberry Fruit Extract Composition Analysis

The total concentration of the anthocyanins compounds in the bilberry fruit extract was 4481.62 mg/L. The most abundant anthocyanins in the extract was delphinidin 3-galactoside with 14.16% in regard to the total amount of the anthocyanins. Table 1 presents the detailed qualitative and quantitative composition of the anthocyanins compounds from the bilberry fruit extract.

### 3.2. The Impact of CCl_4_ and the Bilberry Fruit Extract on the Biochemical Markers of Liver Damage

The liver damage was estimated on the basis of the biochemical markers (GLDH, SDH, and MDH) in the serum (Table 2). The administration of CCl_4_ caused significantly increased activity of the GLDH, SDH, and MDH (*p* < 0.001) enzymes in the serum after 24 h in comparison to the untreated group. The pre-treatment with the anthocyanins from the bilberry extract (group IV) significantly decreased the activity of the biochemical markers of the liver damage (*p* < 0.001) when compared to the results obtained from the animals exposed to the toxic effects of CCl_4_ (group III) exclusively.

### 3.3. The Impact of CCl_4_ and the Bilberry Fruit Extract on the Total Oxidative Status (TOS) and Total Antioxidant Capacity (TAC)

In toxic Group III, CCl_4_ induced an increase in the TOS concentration (*p* < 0.001) with a decrease in the TAC (*p* < 0.001), compared to the untreated group (Table 3). The application of the *Vaccinium myrtillus* fruit extract in Group IV significantly reduced the concentration of the TOC (*p* < 0.01) and increased the concentration of the TAS (*p* < 0.001) in relation to the toxic Group III. The *Vaccinium myrtillus* fruit extract in Group II led to a statistically significant increase in the TAC concentration compared to the untreated Group I.

### 3.4. The Impact of CCl_4_ and the Bilberry Fruit Extract on Pro-Oxidative Markers

We evaluated pro-oxidative markers through the concentrations of CD, LOOH, TBARS, AOPP, H_2_O_2_, GSSG, and activity of NADPH oxidase in the liver homogenate (Table 4; Table 5). The acute exposure to CCl_4_ in Group III caused a statistically significant increase in the concentration of CD, LOOH, TBARS, AOPP, H_2_O_2_, and GSSG, as well as the activities of the NADPH oxidase (*p* < 0.001), compared to the untreated group. A preventive treatment of the anthocyanins components from the bilberry fruit extract in Group IV led to a significant reduction in the concentration of CD, TBARS, H_2_O_2_ (*p* < 0.001), LOOH (*p* < 0.01), GSSG, AOPP (*p* < 0.05), and the activities of the NADPH oxidase (*p* < 0.001), compared to Group III.

### 3.5. The Impact of CCl_4_ and the Bilberry Fruit Extract on Pro-Inflammatory Mediators

Pro-inflammatory mediators in the liver homogenate were evaluated based on the changes in the concentrations of TNF-α, IL-6, NO_2_^−^, NGAL, and CD68, as well as the activity of MPO (Table 6). The acute exposure to CCl_4_ caused a statistically significant increase in the value of pro-inflammatory markers in the liver when compared to the untreated group (*p* < 0.001). A preventive application of the anthocyanins from the bilberry fruit extract led to a significant reduction in the concentrations of TNF-α, IL-6, NO_2_^−^, CD68 (*p* < 0.001), and NGAL (*p* < 0.01), as well as the activity of MPO (*p* < 0.001), compared to Group III.

### 3.6. The Impact of CCl_4_ and Bilberry Fruit Extract on the Arginine and Polyamine Catabolism

The arginine and polyamine catabolism were examined through the arginase, PutOX, and SpmOX activity, as well as the concentration of citrulline, H_2_O_2_, IL-4, and IL-13 (Table 7 and Table 8). The activity of the arginase was statistically significantly increased in the serum and decreased in the liver homogenate in the CCl_4_ group, in comparison to the untreated group (*p* < 0.001). Also, the treatment with CCl_4_ (Group III) caused a statistically significant increase in the activity of PutOX, SpmOX, and the concentration of citrulline and H_2_O_2_, as well as a decreased concentration of IL-4 and IL-13 in the liver when compared to the untreated Group I (*p* < 0.001). The pre-treatment with the anthocyanins (Group IV) led to significantly decreased activity of the arginase in the serum, and concentration of PutOX, SpmOX, H_2_O_2_ (*p* < 0.001), and citrulline (*p* < 0.01) in the liver compared to Group III (CCl_4_). It also resulted in increased activity of the arginase (*p* < 0.001), and the concentration of IL-4 and IL-13 (*p* < 0.01) in the liver, when compared to CCl_4_.

### 3.7. The Impact of CCl_4_ and the Bilberry Fruit Extract on Antioxidative Liver Markers

The antioxidative liver markers were estimated through the changes detected in the activities of the hepatic antioxidant enzymes (GST and QR), as well as through the changes in the concentration of GSH in the liver homogenate (Table 9). The acute poisoning of the animals with CCl_4_ after 24 h (Group III) resulted in a statistically significant decrease of the activity of GST (*p* < 0.001) and QR (*p* < 0.01), as well as of the concentration of GSH (*p* < 0.001) in comparison to the results obtained from Group I. The preventive use of the anthocyanins compounds from the bilberry fruit extract (Group IV) caused a significant increase in the activity of GST (*p* < 0.001), QR (*p* < 0.05), and concentration of GSH (*p* < 0.001), when compared to the results provided by the toxic group (Group III). The treatment with the bilberry fruit extract (treated Group II) induced the activity of GST, QR (*p* < 0.01) and increased the concentration of GSH (*p* < 0.05), in comparison to the results obtained from the untreated Group I.

### 3.8. The Impact of CCl_4_ and Bilberry Fruit Extract on the Histopathological and Immunohistochemical Analyses

A normal histology of the lobular liver tissue was evident in Groups I and II. The toxic effects of CCl_4_ (Group III) caused massive central, perivenular (zone III), and intermedial (zone II) hemorrhagic coagulation necroses of the liver lobules, compared to the untreated Group I (*p* < 0.001) (Figure 1). Acute administration of CCl_4_ at a dose of (3 mL/kg) induced the appearance of severe degenerative changes (macrovesicular hepatocytes) and/or necrosis accompanied by minor reversible changes (vacuolar hepatocytes) or severe degenerative changes (micro and macrovesicular hepatocytes) in treated animals. 

The mononuclear and granulocyte-leukocyte inflammatory infiltrate was detected in all three zones. Hyperplasia of the Kupffer cells was also noticed in comparison to Group I. The capillary micro-hemorrhages were positioned multicentrically in the necrotic and degenerative altered areas. A preventive administration of the anthocyanins from the bilberry fruit extract in Group IV led to the noticeable absence of massive hemorrhagic necrosis (*p* < 0.001), hypoplasia of Kupffer cells, and major degenerative changes in comparison to the CCl_4_ group. In Group III, there were conspicuously present immunohistochemical overexpression pro-inflammatory mediators (TNF-α, COX-2, iNOS, NGAL, and CD68) in comparison to the immunonegative untreated group (Figure 1). The immunohistochemical overexpression was particularly observed in the pericentral and intermediate part of the lobules. The anthocyanins administered in Group IV markedly reduced the immunopositivity of the pro-inflammatory mediators of the liver lobules in regard to the CCl_4_ group.

## 4. Discussion

The liver has a crucial role in the processes of detoxication and metabolism, but it is also liable for damages caused by chemicals, medicines, and environmentally toxic substances, regardless of its powerful regenerative potential [36,37,38]. Carbon tetrachloride is commonly used as the hepatotoxicity model in which the toxic radicals are produced [39]. After it is accumulated in the liver, CCl_4_ is biotransformed in microsomes via the CYP2E1 enzymes to the reactive ^●^CCl_3_ radical. The trichloromethyl radical reacts with oxygen to create a more reactive CCl_3_O_2_^●^ [36,40]. The mechanism of toxic liver damage caused by CCl_4_ occurs upon the induction of oxidative stress, lipid peroxidation, and inflammation. 

During the phase of oxidative stress, the toxic CCl_4_ metabolites lead to the depletion and dysfunction of the antioxidative defense capacities (CAT, SOD, GPx, GST, GR, and GSH) while simultaneously increasing the pro-oxidative markers (Xanthine-oxidase, NADPH oxidase, GSSG, and H_2_O_2_), which results in oxidative stress and liver cell damage [16,35,41,42]. GSSG results from the oxidation of GSH, while NADPH oxidase transfers one electron with NADPH to O_2_, thereby generating ^●^O_2_ˉ [16,27]. GST catalyses the conjugation of GSH and electrophilic toxic substrates, thus creating nontoxic or less toxic and reactive components [43]. GST catalyses a reaction in which GSH neutralizes reactive ^●^CCl_3_, and in that way reduces the degree of the lipid and protein oxidation in the liver cells. Anthocyanins, flavonols, and chlorogenic acids from the bilberry fruit extract, via the activation of the Nrf2/ARE signaling pathway, significantly increase the activity of the antioxidant enzyme of Phase II (GST, QR), as well as the activation of the γ-glutamylcysteine synthetase necessary for the synthesis of GSH [44,45,46]. By inducing the antioxidative defense systems (GST, QR, and GSH), anthocyanins contribute considerably to the prevention of damage to liver cells caused by ROS.

During the phase of lipid peroxidation, the unneutralized toxic CCl_4_ radicals create a covalent link with the proteins and lipids of the membrane of the hepatocyte, mitochondrial, and endoplasmic reticulum, after which the reactive CCl_3_O_2_^●^ radical removes the hydrogen atom from the unsaturated fatty acids of the membranes of the liver cells and organelles, so that the thus created lipid radical induces the lipid peroxidation process which results in the morphological and functional damage in the hepatocytes [16,47]. In the oxidative stress and lipid peroxidation phase, the consumption of antioxidant enzymes leads to a significant reduction in TAC, which in turn causes an increase in TOS [48].

During the inflammatory phase, free radicals CCl_4_ cause the hypertrophy and hyperplasia of the Kupffer cells, which then produce and release numerous harmful and pro-inflammatory components, which results in damage of the liver parenchymatous cells [5,12].

The research presented in this paper demonstrates that the acute toxic effects of CCl_4_ led to the significantly increased activity of the GDH, SDH, and MDH enzymes in comparison to the results of the control-untreated group, which is in accordance with the results obtained in other studies [49,50]. MDH is present in the cytoplasm of the periportal hepatocytes (zone I), GLDH is present in the mitochondria of the centrilobular hepatocytes (zone III), while SDH is present in the cytoplasm and mitochondria of the hepatocytes and represents a specific indicator of the acute hepatocellular damage and necrosis in rats. GLDH and SDH are more specific biomarkers of the hepatocyte damage and necrosis than ALT and AST [51,52]. The toxic metabolites of CCl_4_ led to an increase in the activity of biochemical markers of damage in the serum by inducing the process of lipid peroxidation and by destroying polyunsaturated fatty acids and phospholipids. These processes increased the porosity of the membrane of the hepatocyte, mitochondria, and endoplasmic reticulum, and released the enzymes from the intracellular space into the extracellular space and systemic circulation [3,35]. The pre-treatment with the *Vaccinium myrtillus* fruit extract significantly decreased the serum activities of the specific damage biomarkers (GLDH, SDH), which indicates that the anthocyanins from the bilberry fruit extract are noticeable in protecting the hepatocytes’ membranes and organelles from the toxic effects of CCl_4_. Particularly striking was the decreased activity of MDH, which, combined with the histopathological results, indicated a completely preserved periportal space and slightly damaged centrilobular space, due to the hepatoprotective effects of the bilberry fruit extract.

Acute CCl_4_ poisoning through reactive metabolites ^●^CCl_3_ and CCl_3_O_2_^●^ induced a significant increase in markers of oxidative lipid damage (CD, LOOH, TBARS) and protein (AOPP) in the liver of the rat [16,53,54]. The formation of the CD is an indicator of the initiation of the lipid peroxidation process, where LOOH indicates the progression of the lipid peroxidation process, but also the neutralization of the lipid peroxides by various antioxidant systems, while TBARS is formed by the decomposition of LOOH and denotes the final product in the lipid peroxidation process (Figure 2A) [22,23,55]. AOPP is due to protein damage by the reactive metabolite ^●^CCl_3,_ in which functional inactive proteins are formed [56]. Also, AOPP can activate monocytes involved in the inflammatory reaction through the release of the pro-inflammatory cytokines (IL-1β and TNF-α) [25]. The phenolics-anthocyanins from the bilberry fruit extract led to a significant decrease of the lipid peroxidation level in comparison to the CCl_4_-treated animals, which was accomplished by the termination of the chain process of the lipid peroxidation by adding the hydrogen atom from the hydroxyl (OH) groups of anthocyanins compounds to the unstable and reactive lipid peroxyl radical, which passed into an unreactive and significantly more stable state by forming the lipid hydroperoxides-LOOH. [16,57].

The results of this research confirmed that CCl_4_ caused a significantly increased level of the pro-inflammatory markers (TNF-α, IL-6, MPO, NO, NGAL) and the Kupffer cells identification marker (CD68), as well as immunohistochemical overexpression of pro-inflammatory mediators (TNF-α, COX-2, iNOS, NGAL, and CD68) in comparison to the results obtained from the untreated group, which is in accordance with the results presented in other similar studies [14,29,38,58,59,60]. TNF-α represents a pro-inflammatory cytokine, which has a major role in the pathogenesis of various acute and chronic liver disorders. During the inflammatory phase of the liver damage caused by the carbon tetrachloride, TNF-α contributed to the worsening of the damages provoked by the oxidative stress and inflammation [58]. TNF-α can activate hepatocytes, stellate cells, and endothelial cells, which attract and activate circulating inflammatory cells via released chemokines, which further stimulate the inflammation. On the other hand, TNF-α causes the cells’ necrosis by releasing the elastases and inducing ROS from the Kupffer cells. Moreover, TNF-α stimulates the release of NO from the liver macrophage by the induction of iNOS [49]. The activation and hyperplasia of Kupffer cells has a dual role—it is protective, but also potentially harmful. The protective role is based on the production of hepatoprotective cytokines (IL-10) in the hepatocyte regeneration process, and the harmful role is in the production of pro-inflammatory cytokines which significantly contribute to the pathogenesis of various chronic inflammatory diseases of liver-alcoholic liver disease (ALD), non-alcoholic steatohepatitis (NASH), and non-alcoholic fatty liver disease (NAFLD) [61,62]. In addition to its significant contribution to the process of lipid peroxidation, the trichloromethyl radical has a crucial role in the induction of the inflammation by activating the Kupffer cells in the liver. The activated Kupffer cells then produce and release a variety of inflammatory mediators (TNF-α, IL-6, IL-1β, PGE2, ROS, and eicosanoids), thereby reinforcing the damage of the parenchymatous liver cells [12]. Lipocalin-2 (LCN2) indicates an early biomarker of the liver inflammation, and its values are in correlation with the degree of the liver damage. Acute inflammatory and toxic damage of the liver, as well as the liberated pro-inflammatory cytokines (IL-1beta, IL-6, TNF-alpha) from the activated Kupffer cells, are powerful stimulants that induce the expression of LCN2 from the damaged hepatocytes. The most pronounced immunohistochemical detection of LCN2 is present in the centrilobular zone, which is consistent with the morphological distribution of hepatocyte damage in the liver acinus. On the other hand, the released LCN2 induces the Kupffer cells to release a different range of chemokines that attract the site of toxic damage and inflammation, neutrophils and monocytes [14,63,64]. The Kupffer cells attract neutrophils via CXCL1, CXCL2, CXCL8, and they attract monocytes to the center of inflammation and liver damage via CCL2 and CCL8 [60]. The attracted neutrophils releases ROS and proteases that further aggravate the damage causing necrosis. The increased activity of MPO in the liver homogenate, released from the activated neutrophils, confirms the morphological finding of inflammatory infiltration of neutrophils in the center of damage (Figure 3) [16,65]. The powerful antioxidant properties of anthocyanins, by neutralizing ^●^CCl_3_, reduce excessive activation of the Kupffer cells, which significantly contribute to the anti-inflammatory properties of the bilberry extracts.

An acute administration of CCl_4_ led to a significant increase in Citrulline, PutOx, and SpmOx, as well as a decrease in arginase, IL-4, and IL-13 in the liver homogenate, which was consistent with the results of other researchers [66,67,68]. Citrulline was formed by the conversion of L-arginine via iNOS with NO release. On the other hand, the arginase, stimulated by IL-4 and IL-13, converted L-arginine to L-ornithine. NO reacts with ^●^O_2_ˉ, forming the cytotoxic oxidant-peroxynitrite which induces the reaction of the lipid peroxidation of the membranes and damages DNA [68,69]. Putrescin is formed by the decarboxylation of L-ornithine via ornithine decarboxylase (ODC), while spermidine and spermine are created by the addition of the aminopropyl group via spermidine or spermine synthetase (Figure 2B). The catabolism of putrescine occurs via PutOx in gamma-aminobutyric acid (GABA), until the spermine can be decomposed directly through SpmOx to spermidine. In both catabolic pathways, cytotoxic H_2_O_2_ and acrolein are released [15,70]. Reduced activity of the arginase in the liver after the application of CCl_4_ is explained by extensive enzyme leakage from the damaged hepatocytes into the systemic circulation, by the release of the lysosomal enzymes, and by the decreased concentration of IL-4 and IL-13 [66,68]. Diminished arginase activity, accompanied by an increase of the nitrites in the liver, indicates the diversion of the arginine metabolism towards the citrulline synthesis [71]. Increased PutOx and SpmOx activity are induced by an overwhelming production of ROS (released in the CCl_4_ metabolism) and inflammation through the induction of the NF-ĸB signaling pathway. CCl_4_ provoked the significant induction and overexpression of NF-ĸB [38]. The reduction of polyamines, which contribute significantly to the cell proliferation and regeneration processes, resulted in increased polyamines catabolism through PutOx and SpmOx activity with the production of H_2_O_2_, and diminished the synthesis of polyamines through the inhibitory effect of NO on the process of polyamines synthesis [15]. 

The acute CCl_4_ poisoning caused massive coagulation necroses, followed by minor (vacuolar and hydropic hepatocytes) and major (micro and macrovesicular hepatocytes) degenerative changes, inflammatory infiltrates, and hemorrhaging [5,10,72]. The empirical research conducted in this study shows that the use of highly toxic doses of CCl_4_ (3 mL/kg) caused the massive coagulation necroses and hemorrhage in zones III and II, followed by significant hypertrophy and hyperplasia of the Kupffer cells, when compared to the results obtained from the untreated group [73]. The distribution of the necrotic lesions in the pericentral and intermedial zone corresponded to the high level of the activity of CYP2E1 enzyme, which caused the formation of the toxic metabolites responsible for the hepatocytes’ damage [38].

Regardless of a significantly higher dose of CCl_4_, the Anthocyanins (200 mg/kg) completely prevented necrosis, hyperplasia, and excessive activation of the Kupffer cells, as well as the accumulation of inflammatory infiltrates, which is contrary to the histopathological findings in other studies [10]. In a previous study of Popović et al., anthocyanins (75 mg/kg) did not completely prevent the occurrence of necrosis, which can be explained by the appearance of hyperplasia and hypertrophy of the Kupffer cells, in regard to the CCl_4_ group. 

ROS, released in the CCl_4_ metabolism, can induce the activation of NF-ĸB pathway and cause a significant production of pro-inflammatory mediators (IL-1β, IL-6, TNF-α, iNOS, COX-2, CXCL1, and NGAL), remarkably contributing to the pathogenesis of the liver injury [14,41,74].

The most abundant and active phenolic components in the bilberry extract were anthocyanins, which consisted of a combination of five anthocyanidins (cyanidin, delphinidin, malvidin, peonidin, and petunidin) and three sugar components (galactose, glucose, and arabinose) [10]. Anthocyanidins are different in the position and number of the hydroxyl and methoxy group in the B ring [75]. The pharmacologically most active and prevalent anthocyanidins in the extract were delphinidin and cyanidin, which, due to the presence of three and two hydroxyl groups in the B ring, respectively, provided very strong antioxidant properties [76,77].

## 5. Conclusions

The results obtained in this research undoubtedly confirm the fact that anthocyanins from the *Vaccinium myrtillus* fruit extract evidently decrease the acute hepatotoxic effects caused by CCl_4_. The protective and anti-inflammatory effects are based on the decrease in the catabolism of polyamines and the release of cytotoxic H_2_O_2_, the reduction of the level of lipid peroxidation and pro-oxidative markers, prevention of the arginine metabolism diversion towards the citrulline, reduced activation of the Kupffer cells, restriction of the pro-inflammatory effects of NGAL, and induction of hepatic phase II antioxidant enzymes. This study can be considered as a significant contribution to the possible preventive use of anthocyanins from the extract of bilberry in reducing the hepatotoxic and pro-inflammatory effects of drugs, as well as preventing the development or alleviation of the progression of various chronic liver diseases (e.g., ALD, NASH, and NAFLD).

## Figures and Tables

**Figure 1 antioxidants-08-00451-f001:**
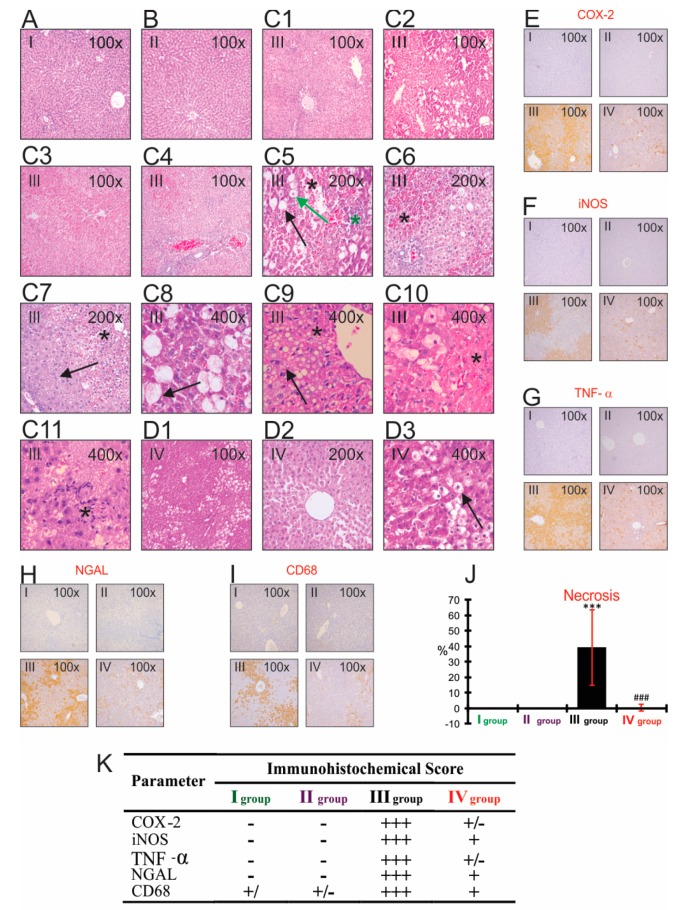
Effect of CCl_4_ and bilberry fruit extract on histopathology of rat liver after 24 h (H&E and Immunohistochemical staining). **I**-group I (untreated group), **II**-group II (treated group), **III**-group III (CCl_4_), **IV**-group IV (treated + CCl_4_). (**A**) group I (untreated group), 100×; (**B**) group II (treated group), 100×; (**C1**–**C11**) group III (CCl_4_); C1-Necrosis followed by minor and severe reversible change, 100×; C2-Necrosis followed by macrovesicular hepatocytes, 100×; C3-Necrosis followed by vacuolar hepatocytes, 100×; C4-Hemorrhagic coagulation necrosis, 100×; C5-Necrosis (🞲), macrovesicular hepatocytes (→), hydropic hepatocytes (→), inflammatory mononuclear infiltrate (🞲), 200×; C6-Hemorrhagic coagulation necrosis (🞲), 200×; C7-Necrosis (🞲), vacuolar hepatocytes (→), 200×; C8- macrovesicular fatty hepatocytes (→), 400×; C9-Necrosis (🞲), microvesicular fatty hepatocytes (→), 400×; C10-Necrosis (🞲), 400×; C11-Inflammatory mononuclear infiltrate (🞲), 400×; (**D1**–**D3**) group IV (treated + CCl_4_); D1-minor reversible change (hydropic hepatocytes), 100×; D2- minor reversible change (vacuolar hepatocytes) and extended sinusoidal spaces, 200×; D3-Hydropic hepatocytes (→), 400×; (**E**) immunohistochemical detection of COX-2, 100×; (**F**) immunohistochemical detection of iNOS (NOS2), 100×; (**G**) immunohistochemical detection of TNF-α, 100×, (**H**) immunohistochemical detection of NGAL, 100×; (**I**) immunohistochemical detection Kupffer cells identification marker-CD68, 100×; (**J**) morphometric analysis extent of the necrotic area on a selected H&E field in the liver. The data show the average value ± S.D. (%) for 10 fields at the magnification of 200× for each animal in each group (H&E staining); (**K**) semiquantitative evaluation of COX-2, iNOS, TNF-α, NGAL, and CD68 immunohistochemistry staining. Staining intensity was graded as: - (negative), +/- (weak positive), + (positive), ++ (strongly positive), or +++ (very strongly positive).

**Figure 2 antioxidants-08-00451-f002:**
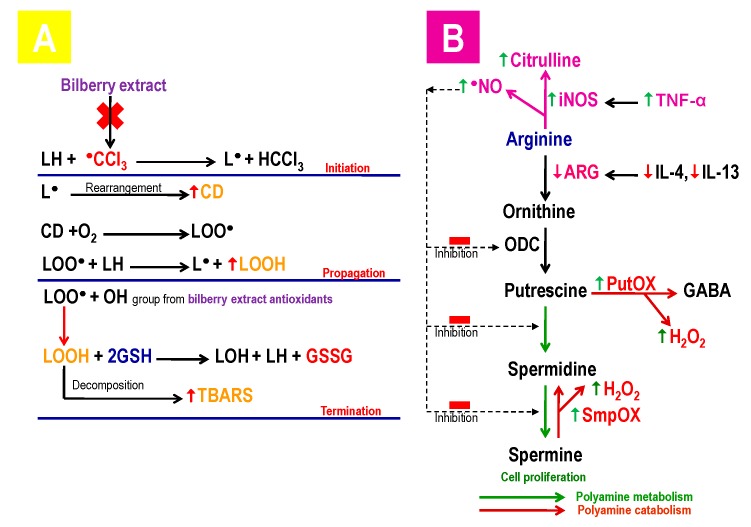
(**A**) The effect of CCl_4_ on lipid peroxidation; (**B**) the role of CCl_4_ on arginine metabolism and polyamine catabolism.

**Figure 3 antioxidants-08-00451-f003:**
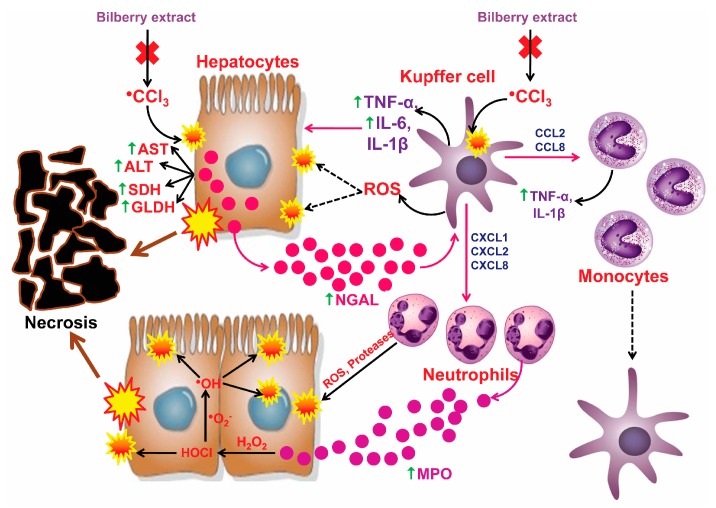
The cellular mechanism of damage to the liver due to acute poisoning of CCl_4_. The effect of toxic metabolite CCl_4_ (^●^CCl_3_) on Hepatocytes and Kupffer cells, and the effect of NGAL on the deterioration of the inflammatory reaction in the liver.

**Table 1 antioxidants-08-00451-t001:** Composition of anthocyanins compounds from bilberry fruit extract.

Individual Anthocyanins in the Bilberry Extract		Concentration in Extract (mg/L)	Percentage of Anthocyanins Compounds in the Extract (%)
**Anthocyanins**	Delphinidin 3-galactoside	634.36	14.16
	Delphinidin 3-glucoside	628.26	14.02
	Delphinidin3-arabinoside	578.08	12.90
	Cyanidin 3-galactoside	497.41	11.10
	Cyanidin 3-glucoside	489.76	10.93
	Cyanidin3-arabinoside	393.90	8.79
	Petunidin 3-glucoside	345.05	7.70
	Petunidin 3-galactoside	169.84	3.79
	Petunidin3-arabinoside	88.73	1.98
	Peonidin 3-glucoside	183.73	4.10
	Peonidin 3-galactoside	43.78	0.98
	Peonidin3-arabinoside	15.55	0.35
	Malvidin 3-glucoside	264.39	5.90
	Malvidin 3-galactoside	89.18	1.99
	Malvidin3-arabinoside	59.60	1.33
	**Total Anthocyanins**	**4481.62**	**100.0**

**Table 2 antioxidants-08-00451-t002:** Effects of CCl_4_ and bilberry fruit extract on biochemical markers of liver injury in rat serum ^a^.

Group	GLDH (U/L)	SDH (U/L)	MDH (U/L)
**I (**not treated**)**	46.16 ± 5.48	31.51 ± 9.49	102.17 ± 28.85
**II (**treated **^b^)**	42.16 ± 5.48	36.29 ± 13.22	96.49 ± 29.46
**III (CCl_4_)**	182.69 ± 17.14 ***	239.23 ± 27.95 ***	310.83 ± 63.28 ***
**IV (**treated **^b^ + CCl_4_)**	91.60 ± 6.51 ^###^	73.89 ± 10.35 ^###^	117.50 ± 14.46 ^###^

^a^ The results expressed as the mean ± S.D. for eight rats in each experimental group; ^b^ Anthocyanins from *Vaccinium myrtillus* (bilberry) fruit extract. *** *p* < 0.001 compared to the untreated group; ^###^
*p* < 0.001 compared to CCl_4_.

**Table 3 antioxidants-08-00451-t003:** Effects of CCl_4_ and bilberry fruit extract on total oxidative status (TOS) and total antioxidant capacity (TAC) in the liver homogenate **^a^**.

Group	TOS (nM/mg Protein)	TAC (nM/mg Protein)
**I (**not treated**)**	4.32 ± 1.16	27.93 ± 2.88
**II (**treated **^b^)**	4.54 ± 1.23	35.44 ± 3.92 ^§§^
**III (CCl_4_)**	16.63 ± 4.19 ***	13.22 ± 2.47 ***
**IV (**treated **^b^ + CCl_4_)**	9.53 ± 1.58 ^##^	23.07 ± 3.81 ^###^

^a^ The results expressed as the mean ± S.D. for eight rats in each experimental group; ^b^ Anthocyanins from *Vaccinium myrtillus* (bilberry) fruit extract. ^§§^
*p* < 0.01 compared to the untreated group; *** *p* < 0.001 compared to the control; ^##^
*p* < 0.01, ^###^
*p* < 0.001 compared to CCl_4_.

**Table 4 antioxidants-08-00451-t004:** Effects of CCl_4_ and bilberry fruit extract on conjugated dienes (CD), lipid hydroperoxide (LOOH), thiobarbituric acid reactive substances (TBARS), and advanced oxidation protein products (AOPP) in the liver homogenate **^a^**.

Group	CD (nM/mg Protein)	LOOH (nM/mg Protein)	TBARS (nM/mg Protein)	AOPP (nM/mg Protein)
**I (**not treated**)**	10.36 ± 1.98	16.18 ± 1.37	2.58 ± 0.56	32.77 ± 4.13
**II (**treated **^b^)**	10.12 ± 1.69	17.04 ± 1.99	2.55 ± 0.41	31.67 ± 3.34
**III (CCl_4_)**	20.23 ± 3.28 ***	34.12 ± 5.22 ***	5.89 ± 0.58 ***	54.68 ± 10.61 ***
**IV (**treated **^b^ + CCl_4_)**	13.98 ± 2.25 ^###^	22.98 ± 3.28 ^##^	4.07 ± 0.81 ^###^	43.9 ± 3.30 ^#^

^a^ The results expressed as the mean ± S.D. for eight rats in each experimental group; ^b^ Anthocyanins from *Vaccinium myrtillus* (bilberry) fruit extract. *** *p* < 0.001 compared to the untreated group; ^#^
*p* < 0.05, ^##^
*p* < 0.01, and ^###^
*p* < 0.001 compared to CCl_4_.

**Table 5 antioxidants-08-00451-t005:** Effects of CCl_4_ and bilberry fruit extract on hydrogen peroxide (H_2_O_2_), oxidized glutathione (GSSG), and NADPH oxidase in the liver homogenate **^a^**.

Group	H_2_O_2_ (nM/mg Protein)	GSSG (nM/mg Protein)	NADPH Oxidase (nM/mg Protein)
**I (**not treated**)**	12.80 ± 2.32	2.51 ± 0.69	12.73 ± 1.43
**II (**treated **^b^)**	12.91 ± 2.89	2.39 ± 0.48	12.51 ± 2.91
**III (CCl_4_)**	27.97 ± 2.95 ***	6.14 ± 0.63 ***	45.69 ± 4.72 ***
**IV (**treated **^b^ + CCl_4_)**	18.89 ± 2.85 ^###^	4.39 ± 1.24 ^#^	23.01 ± 3.15 ^###^

^a^ The results expressed as the mean ± S.D. for eight rats in each experimental group; ^b^ Anthocyanins from *Vaccinium myrtillus* (bilberry) fruit extract. *** *p* < 0.001 compared to the untreated group; ^#^
*p* < 0.05, and ^###^
*p* < 0.001 compared to CCl_4_.

**Table 6 antioxidants-08-00451-t006:** The impact of CCl_4_ and the bilberry fruit extract on pro-inflammatory mediators in liver homogenate **^a^**.

Group	TNF-α (pg/mg Protein)	IL-6 (pg/mg Protein)	Nitrite (μM/mg Protein)	MPO (U/mg Protein)	CD68 (ng/mg Protein)	NGAL (pg/mg Protein)
**I (**not treated**)**	163.56 ± 33.43	15.09 ± 0.78	3.88 ± 0.26	1.07 ± 0.08	3.12 ± 0.69	25.48 ± 10.52
**II (**treated **^b^)**	173.13 ± 35.86	15.42 ± 1.49	4.04 ± 0.39	1.02 ± 0.12	3.06 ± 0.82	23.14 ± 5.92
**III (CCl_4_)**	440.75 ± 54.33 ***	46.27 ± 2.43***	9.03 ± 1.27 ***	2.99 ± 0.33 ***	18.27 ± 4.28 ***	102.21 ± 20.02 ***
**IV (**treated **^b^ + CCl_4_)**	297.89 ± 36.32 ^###^	33.25 ± 2.39 ^###^	5.39 ± 0.84 ^###^	1.68 ± 0.23 ^###^	5.91 ± 1.04 ^###^	58.55 ± 15.34 ^##^

^a^ The results expressed as the mean ± S.D. for eight rats in each experimental group; ^b^ Anthocyanins from *Vaccinium myrtillus* (bilberry) fruit extract. *** *p* < 0.001 compared to the untreated group; ^##^
*p* < 0.01, and ^###^
*p* < 0.001 compared to CCl_4_.

**Table 7 antioxidants-08-00451-t007:** Effects of CCl_4_ and bilberry fruit extract on the metabolism of L-arginine **^a^**.

Group	Arginase ^c^ (μmol/L)	Arginase ^d^ (μmol/mg Protein)	Citrulline ^d^ (μmol/mg Protein)	IL-4 ^d^ (ng/mg Protein)	IL-13 ^d^ (pg/mg Protein)
**I (**not treated**)**	79.58 ± 15.65	0.63 ± 0.07	0.4 ± 0.06	1.14 ± 0.13	35.78 ± 1.78
**II (**treated **^b^)**	83.72 ± 23.38	0.65 ± 0.08	0.39 ± 0.05	1.13 ± 0.18	35.57 ± 1.79
**III (CCl_4_)**	1295.82 ± 282.73 ***	0.20 ± 0.06 ***	0.97 ± 0.19 ***	0.62 ± 0.17 ***	20.41 ± 2.39 ***
**IV (**treated **^b^ + CCl_4_)**	290.26 ± 26.98 ^###^	0.37 ± 0.05 ^###^	0.61 ± 0.06 ^##^	0.87 ± 0.12 ^##^	25.65 ± 2.12 ^##^

^a^ The results expressed as the mean ± S.D. for eight rats in each experimental group; ^b^ Anthocyanins from *Vaccinium myrtillus* (bilberry) fruit extract. ^c^ The parameter was determined in rat serum; ^d^ The parameter was determined in rat liver homogenate; *** *p* < 0.001 compared to the untreated group; ^##^
*p* < 0.01, and ^###^
*p* < 0.001 compared to CCl_4_.

**Table 8 antioxidants-08-00451-t008:** Effects of CCl_4_ and bilberry fruit extract on the catabolism of polyamines in the liver homogenate **^a^**.

Group	PutOX (nM/mg Protein)	SpmOX (nM/mg Protein)
**I (**not treated**)**	3.34 ± 0.84	3.10 ± 0.89
**II (**treated **^b^)**	3.36 ± 0.69	3.58 ± 1.10
**III (CCl_4_)**	7.98 ± 1.49 ***	10.34 ± 1.51 ***
**IV (**treated **^b^ + CCl_4_)**	5.74 ± 0.57 ^##^	7.47 ± 1.09 ^##^

^a^ The results expressed as the mean ± S.D. for eight rats in each experimental group; ^b^ Anthocyanins from *Vaccinium myrtillus* (bilberry) fruit extract. *** *p* < 0.001 compared to the untreated group; ^##^
*p* < 0.01 compared to CCl_4_.

**Table 9 antioxidants-08-00451-t009:** Effects of CCl_4_ and bilberry fruit extract on antioxidative enzymes and reduced glutathione (GSH) in the liver homogenate ^a^.

Group	GST (nM/mg Protein)	QR (nM/mg Protein)	GSH (nM/mg Protein)
**I (**not treated**)**	176.54 ± 17.95	34.22 ± 3.46	58.83.± 5.65
**II (**treated **^b^)**	222.14 ± 19.28 ^§§^	40.54 ± 1.79 ^§§^	66.73 ± 3.09 ^§^
**III (CCl_4_)**	76.59 ± 9.54 ***	24.14 ± 6.63 ***	22.99 ± 5.35 ***
**IV (**treated **^b^ + CCl_4_)**	127.89 ± 16.89 ^###^	33.79 ± 4.96 ^#^	44.42± 7.37 ^###^

^a^ The results expressed as the mean ± S.D. for eight rats in each experimental group; ^b^ Anthocyanins from *Vaccinium myrtillus* (bilberry) fruit extract. ^§^
*p* < 0.05, ^§§^
*p* < 0.01 compared to the untreated group. ** *p* < 0.01, *** *p* < 0.001 compared to the untreated group; ^#^
*p* < 0.05, ^###^
*p* < 0.001 compared to CCl_4_.

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
