# Peer review of "Anthocyanins Protect Hepatocytes against CCl4-Induced Acute Liver Injury in Rats by Inhibiting Pro-inflammatory mediators, Polyamine Catabolism, Lipocalin-2, and Excessive Proliferation of Kupffer Cells"

_antioxidants, 2019, doi:10.3390/antiox8100451_

Round 1
Reviewer 1 Report
The manuscript reveals anthocyanins from bilberry protect CCl4-induced acute liver damage in rat. The experiments were well-designed and data were clearly indicated.
Author Response
RESPONSE TO THE REVIEWERS #1
The authors of this study highly appreciates your well-intentioned and constructive remarks and suggestions.
According to the recommendations and guidelines of the editor, the methodology in the manuscript has been thoroughly and extensively corrected, and changes have been marked in red in the text.
The grammatical and lexical mistakes related to the English language have been corrected by the native English colleagues and experts teaching at our University.
Corresponding author: Dr. Dejan Popović; MD, Mpharm Department of Biochemistry, Faculty of Medicine, University of Niš, Bulevar dr Zorana Đinđića 81, 18000 Niš, Serbia Phone: 00381637195951 e-mail: [email protected]
Reviewer 2 Report
The authors extensively studied the hepatoprotective and anti-inflammatory effects of anthocyanins from the bilberry extract on the acute rat liver injury caused by CCl4. They suggest that the extract decrease the liver damage, pro-oxidative and pro-inflammatory markers, and dissipation of the liver anti-oxidative defense capacities. Their experimental data mainly seems to support many of their arguments. However, some of their suggestions are not based on their findings. This manuscript can be reconsidered for publication only after extensive revision.
Major points:
1. The authors argue that the bilberry extract inhibits activation of NF-kappaB pathway in title, discussion and conclusion. However they did not examine any experiment to investigate this point. Although they found CCl4 induced expression of NF-kappaB in Kupffer cells, translocation of NF-kappaB (p65) into nucleus are inhibited by IkappaB-alpha. Thus at least they should investigate CCl4 induced phosphorylation of IkappaB-alpha or translocation of NF-kappaB (p65) into nucleus with bilberry extract. Furthermore, pro-inflammatory mediators are also increased by activation of various kinase pathways such as ERK, JNK and p38. The authors need to investigate phosphorylation of these kinases.
2. In this study, the authors used rat serum and liver homogenate. They should describe which they used in each table and figure legend.
3. Fig.1A-H should be included in Fig.2 since the authors did not explain these figures in section 3.5, but in section 3.8.
4. The authors should discuss bilberry fruits extract composition and effects of the compounds in discussion section.
Minor point
1. Font size in section 2.2 looks bigger than those in the other section
2. The authors should describe how they defined “necrotic area” more clearly.
3. “NF-kappaB way” had better be described as “NF-kappaB pathway”
Author Response
RESPONSE TO THE REVIEWERS #2
The authors of this study highly appreciate your well-intentioned and constructive remarks and suggestions.
According to the recommendations and guidelines of the editor, the methodology in the manuscript has been thoroughly and extensively corrected, and changes have been marked in red in the text.
Major points:
1.The authors argue that the bilberry extract inhibits activation of NF-kappaB pathway in title, discussion and conclusion. However they did not examine any experiment to investigate this point. Although they found CCl4 induced expression of NF-kappaB in Kupffer cells, translocation of NF-kappaB (p65) into nucleus are inhibited by IkappaB-alpha. Thus at least they should investigate CCl4 induced phosphorylation of IkappaB-alpha or translocation of NF-kappaB (p65) into nucleus with bilberry extract. Furthermore, pro-inflammatory mediators are also increased by activation of various kinase pathways such as ERK, JNK and p38. The authors need to investigate phosphorylation of these kinases.
In this study, the authors have performed the immunohistochemical staining from the paraffin molds of the rat liver tissue sections using NF-κB antibodies. Your remarks and comments are completely justified. However, at this moment, for technical reasons, we are unable to examine the impact of CCl4 and bilberry fruits extract on the IkappaB-alpha (IκBα) and its effect on the NF-kappaB (p65) pathway. In the light of your observation, we will withdraw from the title, results, and discussion the considerations related to the NF-kappaB (p65) pathway. Also, in the discussion, any mention of the NF-kappaB (p65) pathway relates exclusively to the results obtained by other researchers in their studies. The authors of this study would like to thank you for a well-meaning proposal (IκBα, NF-κB, ERK, JNK, and p38 pathways) that they will be able to realize in some future research.
2.In this study, the authors used rat serum and liver homogenate. They should describe which they used in each table and figure legend.
According to your instructions, in the tables and sections in the results, we have specified precisely which parameters were determined in the serum and which were determined in the liver homogenate.
3. Fig.1A-H should be included in Fig.2 since the authors did not explain these figures in section 3.5, but in section 3.8.
According to your guidelines, we have consolidated the histopathological results by merging figure 1 (morphological changes) and part of figure 2 (immunohistochemical results).
4. The authors should discuss bilberry fruits extract composition and effects of the compounds in discussion section. In the discussion, we have highlighted in more detail the influence and pharmacological potential of certain anthocyanidins from the bilberry fruit extract. In particular, we have indicated strong anti-oxidative effects of delphinidin and cyanidin which are the most prevalent anthocyanidins in the extracts.
Minor point
1.Font size in section 2.2 looks bigger than those in the other section.
According to your guidelines, we corrected the technical error in the text and in section 2.2. wealigned the font with the other sections.
2. The authors should describe how they defined “necrotic area” more clearly.
We defined the necrotic field in methodology.
3. “NF-kappaB way” had better be described as “NF-kappaB pathway”
We have adopted your recommendations.
Corresponding author: Dr. Dejan Popović; MD, Mpharm Department of Biochemistry, Faculty of Medicine, University of Niš, Bulevar dr Zorana Đinđića 81, 18000 Niš, Serbia Phone: 00381637195951 e-mail: [email protected]

Reviewer 3 Report
In this manuscript, author investigated the hepatoprotective and anti-inflammatory effects of anthocyanin from the Bilbery extract on the acute liver injury caused by CCl4. Most of results were interesting to understand the hepatoprotective mechanism of Bilbery extract based on inhibition of pro-oxidative mediators and strong anti-inflammatory properties. The data are comprehensive and well presented in the figures, and the experimental approaches appear sound. However, the manuscript needs minor modifications to be considered by Antioxidant.
Minor comments:
Author should seriously consider the name of experimental group to help readers understand better. These name should be simplified and used only one unified name: I(Control) → No treated group, II(BEb) → BEb treated group, III(CCl4) → CCL4+Vehicle treated group, IV(BE+CCl4) → CCL4+BE. Also, the name of each group was maintained the same form in all text and figure. All reagents and instruments should be sequentially described Company, State (Country), Nation. All number should be separated unit. Also, unit should be described same pattern. Author should describe the scientific evidence for setting the single dose of anthocyanins. Author should deposit “Voucher specimens” of Herba Cistanche. Author should justify to use 180 days old of male Wistar rats. Author should justify to analysis COX-2, iNOS, NF-kB, TNF-a, NGAL, CD68 using immunohistochemical detection. ELISA is a better way to quantify it. In Figure 1. a magnification for observation should be added. References should be corrected according to journal guideline. The manuscript should be proof-read by a native English speaker.
Author Response
RESPONSE TO THE REVIEWERS #3
The authors of this study highly appreciate your well-intentioned and constructive remarks and suggestions.
According to the recommendations and guidelines of the editor, the methodology in the manuscript has been thoroughly and extensively corrected, and changes have been marked in red in the text.
1. Author should seriously consider the name of experimental group to help readers understand better. These name should be simplified and used only one unified name: I(Control) → No treated group, II(BEb) → BEb treated group, III(CCl4) → CCL4+Vehicle treated group, IV(BE+CCl4) → CCL4+BE. Also, the name of each group was maintained the same form in all text and figure.
According to your recommendations, we have changed the names of the experimental groups in the methodology, results (tables), and text.
2. All reagents and instruments should be sequentially described Company, State (Country), Nation.
According to your guidelines, we have thoroughly corrected the chemicals and instruments section of the methodology. For each chemical (reagent), its purity, manufacturer (Company) and Country of manufacture were specified in detail. In the methodology, there were specified any type and kind of instruments, Company, and State.
3. All number should be separated unit. Also, unit should be described same pattern.
According to your guidelines, we corrected technical errors that related to the spacing between numbers and units. Also, in all parts of the manuscript the units of the tested parameter are represented in the same pattern.
4. Author should describe the scientific evidence for setting the single dose of anthocyanins.
The dose (200mg/kg b.w.) and the therapeutic regimen (once daily-single dose, 7 days, every day consecutively) of anthocyanins were determined on the basis of previous studies (Popović et al., 2019, Chemico-Biological Interactions; Bao et al., 2008, Journal of Agricultural and Food Chemistry). The anthocyanin dose was administered in a single dose, orally, because any other application (route) of administration would not ensure that all animals received the same anthocyanin dose. For example, it is possible to dissolve the bilberry extract in water and place it in vessels from which rats take in water throughout the day. However, we asserted experimentally that not all animals would take the bilberry solution equally.
5. Author should deposit “Voucher specimens” of Herba Cistanche.
The process of identification of the bilberry fruits was performed by Prof. Nataša Poklar Ulrich at the Department of Food Science and Technology, Biotechnical Faculty, Slovenia. Fresh bilberry fruits were purchased from a commercial company licensed to harvest bilberry fruits in the woods of Koroška and Škofja Loka from the Ministry of Agriculture of the Republic of Slovenia.
6. Author should justify to use 180 days old of male Wistar rats.
The 6 months old rats (180 days) are considered adult rats and are best suited for the inclusion in the experiment. We selected male rats because of the specificity of this study, which considered the anti-inflammatory effects of the bilberry extract. The hormonal status in female rats (estrogen) can significantly affect the inflammatory status and the tested markers. Also, it was found that male rats are more resistant to the effects of toxic chemicals, which significantly reduces the mortality of the animals in the experiment.
7. Author should justify to analysis COX-2, iNOS, NF-kB, TNF-a, NGAL, CD68 using immunohistochemical detection. ELISA is a better way to quantify it.
In this research, the authors used the following Elisa kits: Rat NGAL ELISA kit, Rat CD68 ELISA kit, Rat TNF-α ELISA kit, Rat IL-4 ELISA kit, Rat IL-6 ELISA kit, and Rat IL-13 ELISA kit. We completely agree with you, Elisa kits are a better solution to the quantitative detection of the antigen. However, in the model of CCl4 induced liver injury, it is very interesting to show the histopathological localization of an excessive expression (overexpression) of the antigen. For this purpose, the immunohistochemical detection method has the advantage. In addition, it is possible to easily identify visually the difference in overexpression of antigens in group III (CCl4) and group IV (CCl4 + treated group). In our study, diffuse overexpression was present in zone II and zone III of the liver lobules in the CCl4 group, while focal overexpression in zone II (intermediate) and zone III (centrilobular zone) was present in group IV.
8. In Figure 1. a magnification for observation should be added.
In figure 1, we marked the magnification.
9. References should be corrected according to journal guideline.
We corrected the references.
10. The manuscript should be proof-read by a native English speaker.
The grammatical and lexical mistakes related to the English language have been corrected by the native English colleagues and experts teaching at our University.
Corresponding author: Dr. Dejan Popović; MD, Mpharm Department of Biochemistry, Faculty of Medicine, University of Niš, Bulevar dr Zorana Đinđića 81, 18000 Niš, Serbia Phone: 00381637195951 e-mail: [email protected]

Round 2
Reviewer 2 Report
The authors extensively revised the manuscript. They have responded to many, but not all of the comments on the previous version of the manuscript. There are some points that still need to be addressed.
The author had better spell out “bilberry extract” beside the cartoons of the bilberry in Fig. 2 and 3.
The font of “k” in NF-kB should be symbol font.
In the discussion, we have highlighted in more detail the influence and pharmacological potential of certain anthocyanidins from the bilberry fruit extract. In particular, we have indicated strong anti-oxidative effects of delphinidin and cyanidin which are the most prevalentanthocyanidins in the extracts.
They cited reference 10 in this point in line 607. However, there is no data about antioxidant activity of each compound in it. They should cite appropriate references about antioxidant activity of purified delphinidin and cyanidin there.
Author Response
We would like to thank the Reviewer for her/his helpful comments and suggestions, which will help us to improve the scientific message of our manuscript.
The authors extensively revised the manuscript. They have responded to many, but not all of the comments on the previous version of the manuscript. There are some points that still need to be addressed.
1.The font of “k” in NF-kB should be symbol font.
We have adopted your recommendations in the text of discussion.
2.The author had better spell out “bilberry extract” beside the cartoons of the bilberry in Fig. 2 and 3.
We have adopted your recommendations in figures 2. and 3.
3. In the discussion, we have highlighted in more detail the influence and pharmacological >potential of certain anthocyanidins from the bilberry fruit extract. In particular, we have >indicated strong anti-oxidative effects of delphinidin and cyanidin which are the most >prevalent anthocyanidins in the extracts. They cited reference 10 in this point in line 607. However, there is no data about antioxidant activity of each compound in it. They should cite appropriate references about antioxidant activity of purified delphinidin and cyanidin there.
According to your recommendations, we have removed the inadequate reference and provide a new reference related to the antioxidant properties of purified delphinidin and cyanidin.
